# A Systematic Review of Compensation and Technology-Mediated Strategies to Maintain Older Adults’ Medication Adherence

**DOI:** 10.3390/ijerph20010803

**Published:** 2023-01-01

**Authors:** Hening Pratiwi, Susi Ari Kristina, Anna Wahyuni Widayanti, Yayi Suryo Prabandari, Ikhwan Yuda Kusuma

**Affiliations:** 1Doctoral Program in Pharmacy, Faculty of Pharmacy, Universitas Gadjah Mada, Yogyakarta 55281, Indonesia; 2Department of Pharmacy, Faculty of Health Sciences, Jenderal Soedirman University, Purwokerto 53122, Indonesia; 3Department of Pharmaceutics, Faculty of Pharmacy, Universitas Gadjah Mada, Yogyakarta 55281, Indonesia; 4Department of Health Behavior, Environment, and Social Medicine, Faculty of Medicine, Public Health, and Nursing, Universitas Gadjah Mada, Yogyakarta 55281, Indonesia; 5Pharmacy Study Program, Faculty of Health, Universitas Harapan Bangsa, Purwokerto 53182, Indonesia

**Keywords:** adherence, geriatric, reminder, compensatory, technology

## Abstract

Elderly medication adherence is a challenge in health care. The elderly are often at higher risk for non-adherence, and more likely to be on multiple prescription medications for many comorbidities. This systematic review aimed to explore the current strategies for maintaining older adults’ medication adherence with compensation and technology-mediated strategies. We conducted a systematic review to examine related articles published in the PubMed, Web of Science, and Scopus databases, as well as Google Scholar for additional reference sources by cross-reference review. The Preferred Reporting Items for Systematic Reviews and Meta-Analyses (PRISMA) guidelines were used to guide this review. A total of 217 articles were screened, and 27 studies fulfilled the inclusion criteria. Older adults applied a variety of methods to maintain or enhance their medication adherence. Three studies indicated compensation strategies, 19 studies reported technological assistance, two studies used other strategies (community-offered help or caregivers help), and three studies used a combination of compensation with another strategy or technology. Studies identified various compensation- and technology-based strategies carried out by older adults to help remind them to take medication. This review identified potential benefits of technology and compensation strategy implementation in older adults to increase medication adherence. Although we are conscious of the heterogeneity of the included studies, it remains challenging to determine which elements underpin the most effective approaches.

## 1. Introduction

The World Health Organization defines medication adherence as “the extent to which the individual’s conduct complies with the authorized instructions from a health care professional”. Despite the fact that adherence and compliance are often used similarly, they are not the same thing [1,2]. The degree to which a patient complies with a prescriber’s instructions is known as compliance [1], while “adherence” is used to underline that it is the patient’s decision whether or not to follow the doctor’s recommendations and that doing so should not be a cause for blame [3]. It indicates that adherence is a more positive, proactive action that results in a lifestyle change on the part of the patient, who must adhere to a daily regimen.

Medication adherence is a complicated behavior influenced by many factors including the patient, healthcare professionals, the healthcare system, and individual treatment [1]. Patients’ non-adherence may be intentional or unintentional as a result of these factors. Medication non-adherence can take many different forms, including not having to fill the prescription, not taking it at all, missing a dose, utilizing the incorrect dose, taking it at the wrong time of day, and not taking it as directed (e.g., with or without food). It can also take the form of purposefully stopping the medication for a period of time or stopping it altogether. Intentional non-adherence is the result of patients’ active decision not to take their medication as prescribed; unintentional non-adherence is the result of other factors such as forgetfulness, misunderstanding of medication regimens, access to medication, or language barriers [4].

Elderly patients’ medication adherence is a major challenge in health care. The elderly routinely uses medications to enhance their quality of life, increase their life expectancy, and treat or prevent disease. However, it is apparent that elderly people frequently do not really take their prescriptions as prescribed, which has unfavorable clinical and financial effects [5]. The elderly are more likely to suffer from many comorbidities; they are more likely to develop polypharmacy and have a larger risk of drug non-adherence than the younger population [6,7]. As a result, the patient’s therapeutic benefits are reduced, they have more frequent hospital and physician visits as their medical condition worsens, they spend more money on health care, and they may be overtreated [8]. Because of their prevalent deficiencies in physical dexterity, cognitive skills, and memory, as well as the number of medications they are typically prescribed, elderly people are a particular source of concern [9]. Aging-related physical and cognitive deficits can have a negative impact on adherence. Due to errors in reading prescription labels, discriminating between tablet colors, and opening tablet vials, limitations in visual acuity, hearing, and manual dexterity might have a severe impact on adherence.

Patients’ outcomes may be adversely affected by non-adherence to a therapeutic regimen, which may be even worse in populations with numerous morbidities that require various medication therapies. The elderly are a prime example of this group [10]. For instance, if a patient regularly forgets to take their diabetic medicine, the drug level might not stay within the therapeutic range. Suboptimal glucose management may come from the drug’s inability to effectively remove the excess glucose that is present in the blood when the levels fall below the therapeutic range. If medication adherence is not monitored, the patient’s blood sugar levels may not be optimal when they return for a follow-up appointment, forcing the doctor to adjust the dosage. This needless dosage titration could put the patient at risk for both hypoglycemia and hyperglycemic episodes, which, if poorly managed, could be harmful to the patient [11]. A recent systematic review of direct anticoagulant medications in patients with atrial fibrillation (DOACs; apixaban, dabigatran, edoxaban, and rivaroxaban) in the real-world setting also revealed that around one-third of patients were non-adherent (MPR 80%), which was related to an increased risk of stroke [12].

Several strategies have been developed to improve or maintain patient medication adherence [13]. Patients can use compensation strategies or technology-based strategies. Compensation strategies are typically used proactively to prevent or delay loss of function, and in some situations, they may also serve as an enhanced version of established routines (e.g., organizational habits). The use of a compensation strategy can occur in older adults because of both normal cognitive aging and a neurodegenerative condition that is progressing [14]. Additionally, technology is starting to be used to enhance adherence. Digital interventions for behavior change show potential in studies with current evidence for persons living with chronic disease [15]. Utilizing portable digital devices, mobile health (mHealth) has evolved as a strategy to enhance the implementation of evidence-based medicine and assist public health. It has been investigated whether using this medium can increase treatment adherence [16].

Patients can use a range of adherence aids to help them remember dose times and organize their prescriptions (e.g., medication boxes, alarms) or technology-based strategies. External memory aids, conspicuous locations, and association strategies were all frequently recommended methods, indicating that older persons try to compensate for memory loss [17]. Most published studies have focused more on patient adherence, and few have simultaneously assessed the strategies to maintain or improve patient adherence, particularly in older adults. A prior systematic review conducted by Elizabeth et al. (2009) explored strategies to optimize medication adherence in older adults; however, the results of this review focused entirely on interventional education strategies and memory aids and cues [18]. Chun-Yun Kang (2022) also conducted a systematic review that examined the effectiveness of technology-based intervention in boosting adherence to taking antihypertensive medication. Technology-based interventions reportedly boosted adherence, according to this systematic review [19]. However, this systematic review only addressed technological approaches to managing hypertension medication adherence, whereas our systematic review concentrates primarily on older adults and the strategies adopted by the elderly to maintain and enhance their medication adherence. The authors were interested in conducting a comprehensive review of compensation and technology-mediated strategies to maintain older adults’ medication adherence to fill this gap. This review examines current strategies for maintaining older adults’ medication adherence, especially compensation and technology-mediated strategies. This review sought to answer the following two questions:(1)What compensation strategies could the elderly utilize to maintain or improve their medication adherence?(2)What technology-mediated strategies could the elderly utilize to maintain or improve their medication adherence?

## 2. Methods

### 2.1. Search Strategy

A comprehensive search of databases including PubMed, Web of Science and Scopus was conducted using the keywords “older” AND “drug” AND “reminder” AND “adherence”. We conducted a search on 4 September 2022, and the latest on 6 September 2022, both without limitations on publication time. We also searched Google Scholar for additional reference sources by reviewing the cross-references. The Preferred Reporting Items for Systematic Reviews and Meta-Analyses (PRISMA) enabled the authors to follow guidelines to complete a systematic review [20]. We performed recommended strategies for review; all titles and abstracts were screened to ensure they met the eligibility criteria. Then, we read the full text to determine its relevance. We used Rayyan AI software to avoid duplicate articles. Eligible studies were original articles in the English language. Discussion was used to resolve disagreements among the reviewers (HP, SAK, AWW, YSP, IYK).

### 2.2. Eligibility Criteria

The inclusion and exclusion criteria used in this review are listed in Table 1.

### 2.3. Risk of Bias Assessment

Quality assessments were performed using Joanna Briggs Institute Critical Appraisal Checklists according to the methodological design of the studies: cross-sectional studies, cohort studies, case–control, randomized control trials, and qualitative studies. Any discrepancies between the reviewers (HP, SAK, AWW, YSP, IYK) were resolved through consensus. All included studies were extracted into a summary table (Appendix A). Publication-related information (authors and year), strategies, participants, participant types, duration, measurements, and findings were the primary characteristics extracted.

## 3. Results

The database and citation search generated 217 articles in total, consisting of 76 articles from PubMed, 48 articles from Web of Science, and 83 articles from Scopus. We also searched from Google Scholar for additional reference sources by cross-reference reviews and indicated in the included articles (n = 10).

First, 75 duplicates were eliminated, leaving 142 candidates (databases and citation searching). Second, the titles and abstracts were submitted to the inclusion and exclusion criteria, yielding 46 possibly relevant articles. Most of these articles did not meet the inclusion criteria. Third, all of the documents were thoroughly examined, resulting in 27 articles for a comprehensive review (see Figure 1).

### 3.1. Characteristics of Included Studies

The included studies were published between 2001 and 2022 (See Figure 2) and were conducted in 15 countries: 11 in the United States [17,21,22,23,24,25,26,27,28,29,30], three in Canada [31,32,33], two in China [34,35], and each in Denmark [36], Kenya [37], Taiwan [38], Chile [39], Ghana [40], India [41], Brazil [42], United Kingdom & Mexico [43], Spain [44], Germany [45], Saudi Arabia [46].

Ten studies used randomized control trial (RCT) designs [23,28,32,34,36,38,39,41,44,46], with results generally measured after the intervention, six studies used cross-sectional designs [17,21,30,35,37,43], five studies used mixed-method designs [22,29,31,33,42], two studies used qualitative studies [25,26], two studies used prospective cohort studies [27,40], and two studies used quasi-experimental studies [24,45] (see Appendix A).

### 3.2. Participant Characteristics

All participants were older adults with various characteristics. Thirteen studies reported that the participants were older adults with chronic disease (e.g., hypertension, heart failure, diabetes mellitus) [23,25,28,31,33,34,35,39,40,41,42,44,45], nine studies identified participants as home-living elderly [17,21,24,26,30,32,36,38,43], two studies identified participants as older adults receiving highly active anti-retroviral therapy (ART) [29,37], one was a study of participants receiving buprenorphine treatment [27], one study identified participants as older women with knee osteoarthritis [46], and one study was about older adults with asthma [22].

The sample size varied considerably between studies, ranging from 16 to 21752 participants. We divided the studies into two groups: fewer than 100 older adult participants and above 100 older adult participants. Sixteen studies indicated a sample size of fewer than 100 older adult participants [23,24,25,26,27,29,30,31,32,33,38,41,42,44,45,46], and 11 studies more than 100 participants [17,21,22,28,34,35,36,37,39,40,43]. The studies’ durations varied based on their study design; nevertheless, five studies did not specify their survey’s duration (not explicitly/NE) [17,21,33,37,43] and others ranged from 45 min to a year (see Appendix A).

### 3.3. Types of Strategies

It can be seen in Table 2 that the strategies used by older adults to maintain or improve their medication adherence vary; three studies indicated compensation strategies [17,21,43], 19 studies reported technological assistance [23,24,25,27,28,29,30,31,32,33,35,36,37,39,41,42,44,45,46], two studies used other strategies (i.e., community-offered help or caregivers’ help) [38,40], and three studies used a combination of compensation and other strategy or technology [22,26,34] (See Table 2, Figure 3 and Figure 4).

The most frequent compensation strategy is using external help (alarm clocks, pillboxes). This was discovered by 14% of the included studies, and 11% of studies went on to state that everyday routines, location, and mental awareness are frequently used to enhance adherence (see Figure 4). As much as 31% of the included studies indicate that the application on a smartphone or tablet is the technology-mediated strategy that is most used. Additionally, up to 30% of the studies that were examined stated that text message or phone call reminders assisted the elderly in boosting adherence (see Figure 5).

Boron et al. (2013) surveyed 354 older adults, finding that the location strategy was that most frequently endorsed by the elderly, followed by the visibility and association strategies, the use of a pill caddy, mental planning strategies, and the last physical pain [17]. From the strategies that had been carried out, most of the elderly in this study used compensation strategies and did not require a technology-mediated strategy to manage their medications. On the other hand, an RCT conducted by Goldstein et al. (2014) intended to see if implementing a telehealth intervention (electronic pillbox) or an m-health intervention (smartphone application) to remind older adults with heart failure would enhance medication adherence [23]. According to this study, older adults who used a telehealth device (an electronic pillbox) adhered 80% of the time, whereas older adults who used a smartphone app adhered 76% of the time. Participants preferred the m-health intervention over the telehealth intervention in both scenarios (*p* < 0.001). Participants who received smartphones with the m-health app (mean score 48.7) gave their device a considerably higher rating on patient acceptability and device usefulness than those who received telehealth intervention.

An RCT on the use of the ALICE application was also conducted by Mira et al. (2014). The ALICE tablet-based medication self-management application was created to assist patients in remembering to take all of their prescribed medications at the right times, differentiating between medications to prevent confusion, avoiding known potential interactions and frequent medication usage mistakes, and understanding how to properly store their medications. ALICE was also created to keep in mind the advice of medical professionals regarding dietary and exercise regimens. According to this study, patients in the experimental group scored higher on the Morisky Medication Adherence Scale and reported missing fewer doses of their medications (*p* < 0.001 and *p* = 0.02, respectively) [44]. Swanlund (2010) interviewed and encouraged community-dwelling adults to describe situations or variables that help or hinder cardiovascular medication management processes [26]. This study explained that the strategies used by patients varied and were related to compensation and other strategies such as caregivers’ assistance.

### 3.4. Outcome and Measurement

The findings of the included studies were quite diverse in terms of strategies, and they lacked clearly defined primary outcomes (adherence), implying some degree of comparability among the studies. This was also possible because of the various study designs. Instead, we conducted a narrative analysis of the research as a best-case alternative (See Appendix A).

Four studies on compensation strategies found that the external help strategy was the most used by older adults to remember taking their medication [17,21,22,43]. Brooks et al. (2014) reported that patients who said they kept their medications in specific locations did so most often at their bedside (20.1% of the overall sample), followed by the bathroom (9.2%) [22]. Furthermore, the bathroom and daily routine were stronger predictors of adherence than the demographic and cognitive characteristics that were considered. This is in line with Stawarz et al. (2016), who found that older persons kept their tablets by the bed (52%), in a handbag or purse (19%), in the bathroom (10%), or in a make-up bag (6%); the location (e.g., bathroom) and accompanying goods (e.g., purse) gave additional clues [43].

Another study discussed the Telesvar (TS), an electronic reminder device that is used to evaluate drug adherence-based technology compared with pill counting. The results revealed that the two strategies substantially differed in terms of total adherence rates (TS 79%, pill count 92%). When there were more than three intake times, the TS showed a substantial decrease in adherence. This was not the case with the pill count [36]. Another study evaluated the viability of an eDossete intervention, a technology-mediated medication management device. The average total adherence rate was 82%, according to the findings (49%–100%). This study indicated how the eDosette could help with medication adherence implementation [32].

Measurements from included studies can be seen (Appendix A) to vary depending on the study design. Twenty studies used questionnaires or item scales [17,21,22,24,27,29,30,31,33,34,35,37,38,39,41,42,43,44,45,46], two studies used qualitative analysis for their analysis [25,26], one study used pill counts [36], one used pillbox opening bins and electronic self-reports [23], one used total dose taken [32], one used a modified version of the proportion of days covered [28], and one study used hypertension control (<140/90 mmHg) [40].

## 4. Discussion

### 4.1. Compensation Strategies

In the present study, we provided a comprehensive review of the literature on compensation and technology-mediated strategies to maintain older adults’ adherence. Our results indicated that various compensation strategies are carried out by older adults to help remind them to take their medications. Compensation is a collection of activities aimed at minimizing or responding to loss in general (either actual or perceived). Compensation strategies are usually used in advance to prevent or delay the loss of function, and in some situations, they can also be seen as an exaggeration of long-term habits (e.g., organizational habits). They are intended to increase a person’s capacity to operate in daily life [47].

Our review revealed that older adults use a variety of compensation strategies, such as daily routine, memories, associations, simplification, location, taking medication if necessary, external help tools, visibility, printed medication lists, specific times, visual of compartments, mental awareness, repeating the instruction, education, and physical pain, to maintain or improve their medication adherence. According to research on strategy utilization in the context of medication adherence, older adults used internal (mainly mental associations) and external (physical objects and/or locations) methods to remember their medications. It is critical for older persons to remember to take their medications, not only because it reflects their memory, but also because non-adherence can have a negative impact on their health and independence [17]. Our findings concluded that external reminders were among the most often reported and were related to stronger self-reported adherence and memory self-efficacy. The presence of auditory and visual signals may be connected to successful medication adherence. A consistent practice, such as taking medicine at the same time and in the same place every day, can also help with adherence [17].

Several studies have investigated how people remember to take their medications and how they manage them. Older adults attempt to limit their risks of making a mistake or the consequences [48]. Compensation strategies include taking medications out the night before so they are not forgotten in the morning, turning pill bottles over to indicate that the medication has been taken, and leaving medications in meaningful places where they will be encountered at the appropriate time [49]. Another previous study found that older persons use a range of compensation methods to help them remember to take their medications, with external reminders (such as storing pills in a conspicuous position) being among the most prevalent [47].

We discovered a considerable emphasis on memorized routines that older persons rely on to maintain adherence, with times, locations, and specific events playing important roles in adherence. Brooks et al. (2014) reported that asthma geriatrics who kept their medicine in the bathroom or included it in their daily routine were more likely to be adherent than those who utilized other methods [22]. A small percentage of people utilized these strategies, but they were more likely to adhere than those who used the more prevalent strategy of keeping drugs in the bedroom, including near the bedside.

### 4.2. Technology-Based Strategies

#### 4.2.1. Text Message Reminders

Our findings suggested that the text message strategy is one that older adults use to remember their medications. Short messaging service (SMS), sometimes known as text messaging, is one of the most widely used interpersonal mobile communication channels. It involves the creation and real-time transmission of alphanumeric messages of 160 characters or less [50]. However, due to the availability of free mobile-messaging software such as WhatsApp, Telegram, and Facebook Messenger, texting usage has decreased marginally in recent years.

A text message can help improve healthcare services and serve as a reminder (e.g., for daily medication adherence). Individuals who require additional support or structure to recall facts or engage in a habit may benefit from text messaging [51]. Because of their automated administration, long-standing use, and convenience of use, text-message-based interventions have been rated viable and beneficial. Mobile technology may be an effective way to increase medication adherence, given its expanding global usage and appeal among older persons [16].

Text message reminders are available on any existing mobile device and can be utilized by people of various socioeconomic backgrounds and ages. Text message-based interventions are frequently used to organize prescription cautions and reminders to enhance medication adherence. This strategy provides self-management support for patients seeking to improve health behaviors, particularly older persons with various chronic conditions, as a potentially low-cost method [52].

Our review found that participants in one of the included studies thought text message reminders were a practical, simple, and adaptable technique for developing a medication-taking practice. Older adults are active and capable users of mobile technology. Medication adherence is seen as more accessible with text messaging and mobile phone applications [25]. One of the studies in our review included an RCT with a mobile phone text message intervention to encourage African American adult HIV-positive patients’ adherence to ART and co-occurring chronic disease drugs. In contrast to self-efficacy or affect scores, individuals reported statistically significant improvements in medication adherence after 8 weeks [29]. This was also discussed by Maharani et al. (2018), who noted that patients with chronic conditions such as hypertension have been shown to respond well to SMS and phone calls, especially when it comes to adherence to medicines, exercise, and food, but these reminders are only helpful for a limited time [53]. While unidirectional text messages may aid adherence by reminding patients when to take their medications, they are unlikely to help with other frequent pharmaceutical issues such as difficulty understanding medication instructions, payment issues, or difficulties coordinating a multi-drug regimen. Patients are likely to require direct assistance from the clinical care team to solve these more difficult issues.

#### 4.2.2. Electronic Pillbox

The pillbox is the most often used device for promoting medication adherence. People can manage their medicines on their own, double-check whether they have taken them, eliminate the risk of taking them twice or not and reduce medication errors [54]. Pillboxes were created to help the elderly manage their medications. However, using them can lead to mistakes, especially when dealing with complicated medication regimens. When standard pill organizers are not enough, some improvement can be made by using pillboxes with an alarm clock to keep prescription schedules on track [55]. Medication event monitoring systems (MEMSs), often known as electronic pillboxes, are available. They offer the added function of using alarms to remind patients to take their medications, and they are regarded as the gold standard for determining adherence [56].

Pillboxes do exist; however, most of them have limited functionality, are unsuitable for the elderly, or are too large to carry around. To create a truly functional smart pillbox, the latest sweeping smart technologies need to be simply incorporated. At the same time, its ease of use has to be appropriate for the elderly to apply, due to their limited knowledge and experience [57]. Hayes et al. (2006) proposed the MedTracker, an electronic pillbox, as a valuable tool in investigations of treatment effects on medication management in the elderly. This technology improves on existing systems by enabling mobility, frequent and automatic data gathering, more specific information regarding non-adherence and medication problems, and a 7 day drug store pillbox interface [58].

In accordance with our findings, electronic pillboxes are highly helpful for older adults to take their medications, but their use necessitates instruction, especially during the initial use. An included study conducted by Goldstein et al. (2014) divided participants into four groups, i.e., pillbox silent, pillbox reminding, smartphone silent, and smartphone reminding. The pillboxes were either programmed with reminder alarms for the times patients were supposed to take each of four monitored medications, or they were not, making the device a passive adherence monitor. Despite the findings of adherence to the electronic pillbox in this study being higher than that to a smartphone application, people who were given the smartphone application rated their device higher than people given the telehealth medicine container (electronic pillbox). Some of them found the telehealth medication containers’ reminder features bothersome. Participants likely opened bins in the active state simply to silence the alarms, thereby recording a medication-taking event, regardless of whether a tablet was actually taken [23].

#### 4.2.3. Smartphone Mobile Applications

People use smartphone applications for banking, online shopping, home security, social networking, and other daily activities. Those who used mobile applications less frequently nevertheless have social networking (e.g., Facebook), fitness tracking apps, and games on their phones. While application features such as drug interaction information and health indicator tracking were not directly related to medication adherence, people expressed a strong desire to use the applications to validate and track their health observations and symptoms and speculated that this increased motivation to use the applications would lead to increased use of them as medication reminders [56].

One of included studies found that participants highlighted the additional advantages of applications that included capabilities of greater interactivity, individualized health monitoring, and personalized medication information (e.g., side effects, drug–drug interactions). A patient and provider interface was desired, allowing medical providers to follow medication adherence with the patient at the same time and, ideally, discover non-adherence quickly. Many users saw the applications as instruments for generating and forming a habit of sticking to their prescription regimens and assisting in developing better health practices [25].

One study found a smartphone-based medication self-management system with a real-time alert feature to improve medication adherence [59]. The system under investigation had two important characteristics. Patients’ medication histories were stored and maintained in an accurate and accessible fashion. Second, the system sent an electronic reminder if a medication was missed. Prescription data were input together with photos of the drug, which were read using barcodes with a smartphone camera.

The capacity of a smartphone application to customize the intervention with the patient and readily incorporate co-interventions such as counseling, reminders, and motivational and educational text messages is the reason why mobile smartphone applications were successful in improving medication adherence in our review. We believe that there is a significant opportunity to develop medication adherence applications that will benefit elders.

#### 4.2.4. Limitations of This Review

There is a limitation to this review. The difficulty in compiling the results was because of the vast heterogeneity of methods and results from the articles found. Therefore, it remains challenging to determine which elements truly underpinned the effective strategies. A meta-analysis approach is the most evidence-based and comprehensive method of summarizing empirical findings. Therefore, it would be beneficial for future research.

## 5. Conclusions

This review found that adherence strategies such as compensation and technology-mediated strategies have been demonstrated to help the elderly remember older adults’ medications. Older adults use a variety of compensation strategies, such as daily routines, memories, associations, simplification, location, taking medication if necessary, external help tools, visibility, printed medication lists, specific times, visual of compartments, mental awareness, repeating the instructions, education, and physical pain, to maintain or improve their medication adherence. Additionally, technology-based strategies including the use of electronic reminders, text message reminders, tablet/smartphone applications, and electronic pillboxes significantly help the elderly manage their medication adherence.

## Figures and Tables

**Figure 1 ijerph-20-00803-f001:**
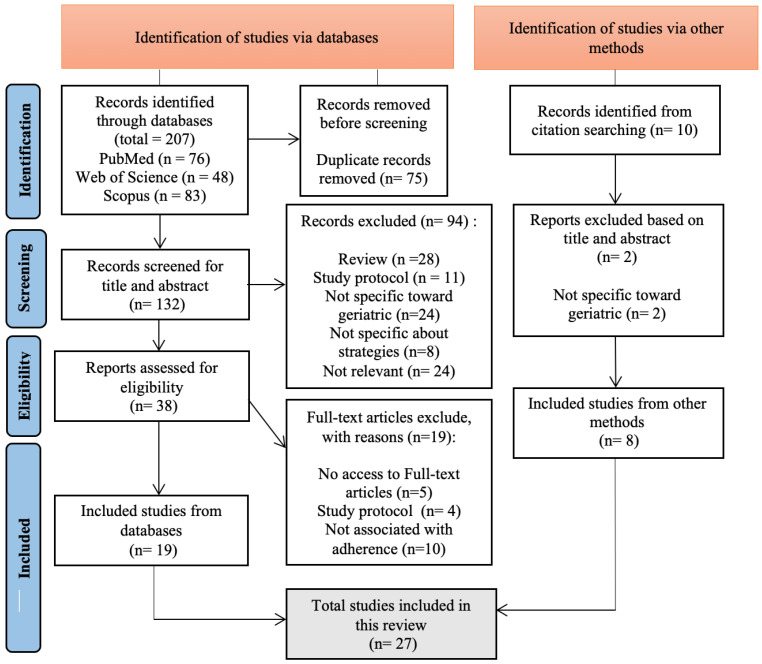
PRISMA diagram of this review.

**Figure 2 ijerph-20-00803-f002:**
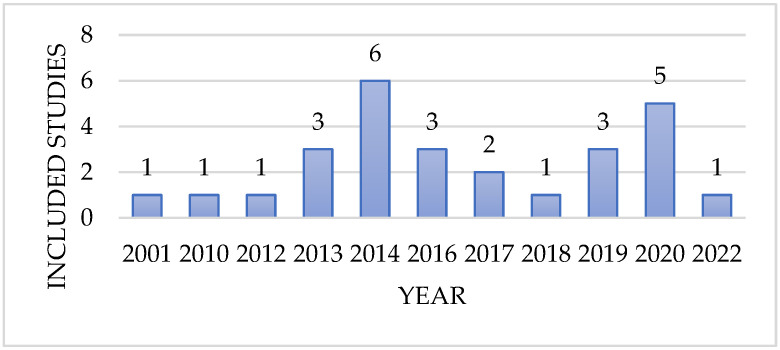
The year range of included studies.

**Figure 3 ijerph-20-00803-f003:**
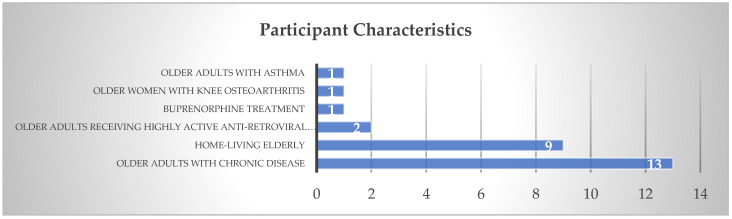
Participant Characteristics.

**Figure 4 ijerph-20-00803-f004:**
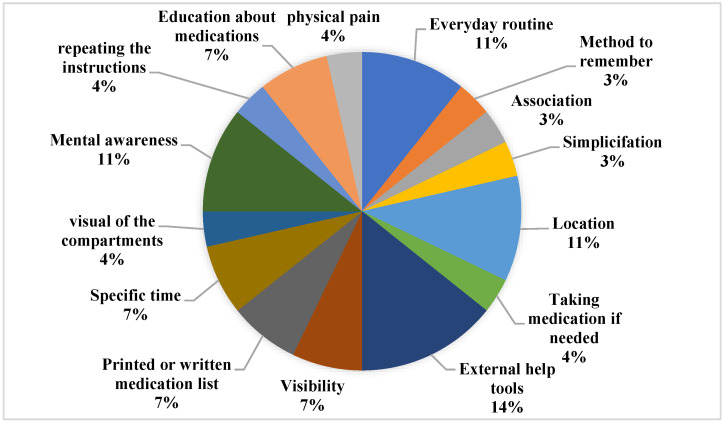
Compensation strategies of the included studies.

**Figure 5 ijerph-20-00803-f005:**
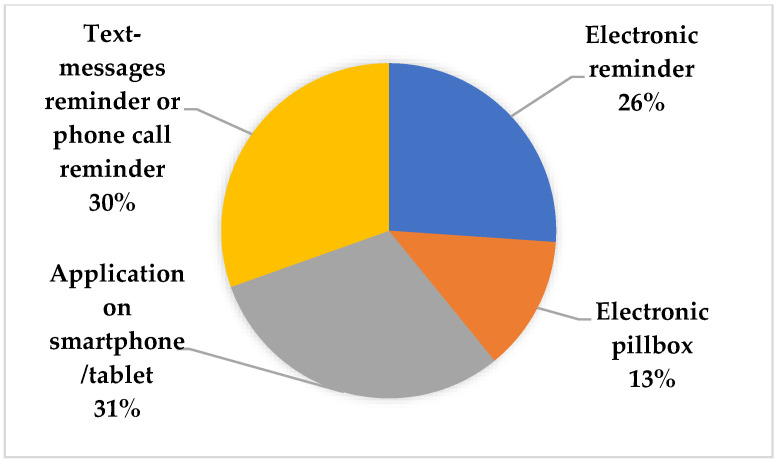
Technology-mediated strategies of the included studies.

**Table 1 ijerph-20-00803-t001:** Inclusion and Exclusion Criteria.

Included	Excluded
Original article	Review articles
Full-text available	Commentary articles
Study design: Randomized control trial Cross-sectional, case–control, cohort Qualitative study Mixed-method	Handbooks or guidelines
Associated with older adults’ medication adherence	Study protocol
Focused on strategies (compensation or technology-mediated strategies) used by older adults to manage their medications.	

**Table 2 ijerph-20-00803-t002:** Types of strategies used to enhance medication adherence.

Strategy	Branin et al.	Swanlund et al.	Harbig et al.	Boron et al.	Kinyua et al.	Wang et al.	Brooks et al.	Vollmer et al.	Goldstein et al.	Ligons et al.	Stawarz et al.	Varleta et al.	Tofighi et al.	Adler et al.
**Compensation**														
1. Everyday routine	✓	✓					✓							
2. Method to remember	✓													
3. Association	✓													
4. Simplification		✓												
5. Location				✓			✓				✓			
6. Taking medication if needed							✓							
7. External help tools (i.e., pillbox, alarm clocks)	✓			✓			✓				✓			
8. Visibility		✓		✓										
9. Printed or written medication list	✓						✓							
10. Specific time of day							✓				✓			
11. Visual of the compartments by color, shape, and size.		✓												
12. Mental awareness	✓	✓		✓										
13. Repeating the instructions to oneself more than one time	✓													
14. Education about medications		✓												
15. Physical pain				✓										
**Technology-mediated strategy**														
1. Electronic reminder (i.e., device)			✓							✓				
2. Electronic pillbox									✓					
3. Application on smartphone/tablet									✓					
4. Text-message reminder or phone call reminder					✓			✓				✓	✓	
**Other strategies**														
1. Community-offered help						✓								✓
**Strategy**	**Shen et al.**	**Siu et al.**	**Park et al.**		**Raj et al.**	**Vieira et al.**	**Mao et al.**	**Mira et al.**	**Grindrod et al.**	**Mertens et al.**	**Dasgupta et al.**	**Pagan et al.**	**Patel et al.**	**Alasfour et al.**
**Compensation**														
1. Everyday routine														
2. Method to remember														
3. Association														
4. Simplification														
5. Location														
6. Taking medication if needed														
7. External help tools (i.e., pillbox, alarm clocks)														
8. Visibility														
9. Printed or written medication list														
10. Specific time of day														
11. Visual of the compartments by color, shape, and size.														
12. Mental awareness														
13. Repeating the instructions to oneself more than one time														
14. Education about medications	✓													
15. Physical pain														
**Technology-mediated strategy**														
1. Electronic reminder (i.e., device)		✓				✓	✓						✓	
2. Electronic pillbox	✓												✓	
3. Application on smartphone/tablet			✓					✓	✓	✓	✓			✓
4. Text-messages reminder or phone call reminder			✓		✓							✓		
**Other strategies**														
1. Community-offered help														
2. Caregivers														

## Data Availability

Data supporting the results can be found in Appendix A.

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
