# Peer review of "A Systematic Review of Compensation and Technology-Mediated Strategies to Maintain Older Adults’ Medication Adherence"

_ijerph, 2023, doi:10.3390/ijerph20010803_

Round 1
Reviewer 1 Report
The authors have addressed a relevant topic, which doesn't appear to have been recently examined in a review. The manuscript could further benefit from 1) a clear distinction between compliance and adherence - it is commendable that the authors focused on the active healthy behavior rather than the passive submission to medical instructions (compliance). This notion can be further underlined in the introduction. Potentially the authors could also discuss whether other studies have focused on the use of digital technologies for the improvement of compliance to medications; 2) some background information or some discussion around the health menaces of limited adherence to medication (for instance, a recent study analysed the increase in mortality risk among atrial fibrillation patients receiving more than 4 medicines on a daily basis).
An additional proof - read to correct minor spelling issues would also be necessary.
Reviewer 2 Report
The authors present a systematic literature review of studies aimed at improving medication adherence strategies of older adults. The paper is very interesting and provides valuable knowledge on the subject. I have a few recommendations for improvement.
INTRODUCTION
- What were the research questions for the systematic review?
METHODS
- I think eligibility criteria are still quite vague. There are a number of studies, especially from the digital health literature that are concerned with design and development of apps with older adults. Were these included or excluded? Did authors include qualitative studies that investigate compensation strategies?
- Surprisingly, the review includes only 3-4 studies from recent years. Authors should consider including a bar chart to show number of publications in different years for each stage of the screening process.
- The article selection does not seem very robust. I do notice that some relevant studies (particularly, those related to digital health) that meet the eligibility criteria have not been included, e.g.
Dasgupta D, Johnson RA, Chaudhry B, Reeves KG, Willaert P, Chawla NV. Design and Evaluation of a Medication Adherence Application with Communication for Seniors in Independent Living Communities. AMIA Annu Symp Proc. 2017 Feb 10;2016:480-489. PMID: 28269843; PMCID: PMC5333254.
Mira J, Navarro I, Botella F, et al. A Spanish pillbox app for elderly patients taking multiple medications: randomized controlled trial. Journal of medical Internet research. 2014;16(4):e99.
Grindrod KA, Li M, Gates A. Evaluating user perceptions of mobile medication management applications with older adults: a usability study. JMIR mHealth and uHealth. 2014;2(1)
RESULTS
- Authors should include a diagram that help conceptualize the overall state of the art in terms of strategies and outcomes of the interventions.
- The authors should include some charts/graphs to help visualize the characteristics of all the studies.
DISCUSSION
- This is fine, however, without the research questions, I do not know why am I reading this section and what am I supposed to get out of it.
CONCLUSION
- The final sentence of the conclusion, although is quite reasonable, it is not based on the results of the systematic review. In fact, it seems quite disconnected from the rest of the paper. How do we come to this conclusion?
